# DIFFERENTIABLE EULER CHARACTERISTIC TRANSFORMS FOR SHAPE CLASSIFICATION

**Ernst Röell**[1,2]**, Bastian Rieck**[1,2]

[1]AIDOS Lab, Institute of AI for Health, Helmholtz Munich
[2]Technical University of Munich (TUM)

## ABSTRACT

The *Euler Characteristic Transform* (ECT) is a powerful invariant, combining geometrical and topological characteristics of shapes and graphs. However, the ECT was hitherto unable to learn task-specific representations. We overcome this issue and develop a novel computational layer that enables learning the ECT in an end-to-end fashion. Our method, the *Differentiable Euler Characteristic Transform* (DECT) is fast and computationally efficient, while exhibiting performance on a par with more complex models in both graph and point cloud classification tasks. Moreover, we show that this seemingly simple statistic provides the same topological expressivity as more complex topological deep learning layers.

## 1 INTRODUCTION

Geometrical and topological characteristics play an integral role in the classification of complex shapes. Regardless of whether they are represented as point clouds, meshes (simplicial complexes), or graphs, the multi-scale perspective provided by methods from *topological data analysis* (TDA) can be applied effectively for classification tasks. Of particular relevance in this context are the *Persistent Homology Transform* (PHT) and the *Euler Characteristic Transform* (ECT). Originally introduced by Turner et al. (2014), recent work proved under which conditions both transforms are invertible, thus constituting an injective map (Crawford et al., 2020; Ghrist et al., 2018). Both transforms are based on the idea of looking at a shape from multiple directions, and evaluating a multi-scale topological descriptor for each such direction. For the PHT, this descriptor is *persistent homology*, a method for assigning multi-scale topological features to input data, whereas for the ECT, the descriptor consists of the *Euler characteristic*, an alternating sum of so-called simplices in a topological space. The collection of all these direction–descriptor pairs is then used to provide a classification or solve an optimisation task. This approach is mathematically sound, but evaluating *all* possible directions is infeasible in practice, thus posing a severe limitation of the applicability of the method.

**Our contributions.** We overcome the computational limitations and present a *differentiable, end-to-end-trainable Euler Characteristic Transform*. Our method (i) is highly scalable, (ii) affords an integration into deep neural networks (as a layer or loss term), and (iii) exhibits advantageous performance in different shape classification tasks for various modalities, including graphs, point clouds, and meshes.

## 2 RELATED WORK

We first provide a brief overview of *topological data analysis* (TDA) before discussing alternative approaches for shape classification. TDA aims to apply tools from algebraic topology to data science questions; this is typically accomplished by computing algebraic invariants that characterise the *connectivity* of data. The flagship algorithm of TDA is *persistent homology* (PH), which extracts multi-scale connectivity information about connected components, loops, and voids from point clouds, graphs, and other data types (Barannikov, 1994; Edelsbrunner & Harer, 2010). It is specifically advantageous because of its robustness properties (Skraba & Turner, 2020), providing a rigorous

approach towards analysing high-dimensional data. PH has thus been instrumental for shape analysis and classification, both with kernel-based methods (Reininghaus et al., 2015) and with deep neural networks (Hofer et al., 2017). Recent work even showed that despite its seemingly discrete formulation, PH is differentiable under mild conditions (Carrière et al., 2021; Hofer et al., 2019; 2020; Moor et al., 2020), thus permitting integrations into standard machine learning workflows. Of particular relevance for shape analysis is the work by Turner et al. (2014), which showed that a transformation based on PH provides an injective characterisation of shapes. This transformation, like PH itself, suffers from computational limitations that preclude its application to large-scale data sets. As a seemingly less expressive alternative, Turner et al. (2014) thus introduced the *Euler Characteristic Transform* (ECT), which is highly efficient and has proven its utility in subsequent applications (Amézquita et al., 2021; Crawford et al., 2020; Marsh et al., 2022; Nadimpalli et al., 2023); see Munch (2023) for an overview. It turns out that despite its apparent simplicity, the ECT is also injective, thus theoretically providing an efficient way to characterise shapes (Ghrist et al., 2018). A gainful use in the context of deep learning was not attempted so far, however, with the ECT and its variants (Jiang et al., 2020; Kirveslahti & Mukherjee, 2023) still being used as *static* feature descriptors that require domain-specific hyperparameter choices. **By contrast, our approach makes the ECT end-to-end trainable, resulting in an efficient and effective shape descriptor that can be integrated into deep learning models**. Subsequently, we demonstrate such integrations both on the level of *loss terms* as well as on the level of *novel computational layers*.

In a machine learning context, the choice of model is typically dictated by the type of data. For *point clouds*, a recent survey (Guo et al., 2021) outlines a plethora of models for point cloud analysis tasks like classification, many of them being based on learning equivariant functions (Zaheer et al., 2017). When additional structure is being present in the form of graphs or meshes, *graph neural networks* (GNNs) are typically employed for classification tasks (Zhou et al., 2020), with some methods being capable to either learn *explicitly* on such higher-order domains (Bodnar et al., 2021a;b; Ebli et al., 2020; Hacker, 2020; Hajij et al., 2020) or harness their topological features (Horn et al., 2022; Papillon et al., 2023).

## 3 MATHEMATICAL BACKGROUND

Prior to discussing our method and its implementation, we provide a self-contained description to the *Euler Characteristic Transform* (ECT). The ECT often relies on *simplicial complexes*, the central building blocks in algebraic topology, which are used extensively for calculating homology groups and proving a variety of properties of topological spaces. While numerous variants of simplicial complexes exist, we will focus on those that are embedded in $\mathbb{R}^n$. Generally, simplicial complexes are obtained from a set of points, to which higher-order elements—*simplices*—such as lines, triangles, or tetrahedra, are added inductively. A $d$-simplex $\sigma$ consists of $d+1$ vertices, denoted by $\sigma = (v_0, \dots, v_d)$. A $d$-dimensional simplicial complex $K$ contains simplices up to dimension $d$ and is characterised by the properties that (i) each face $\tau \subseteq \sigma$ of a simplex $\sigma$ in $K$ is also in $K$, and (ii) the non-empty intersection of two simplices is a face of both. Simplicial complexes arise 'naturally' when modelling data; for instance, *3D meshes* are examples of 2-dimensional simplicial complexes, with $0$-dimensional simplices being the vertices, the $1$-dimensional simplices the edges, and $2$-dimensional simplices the faces; likewise, *geometric graphs*, i.e. graphs with additional node coordinates, can be considered $1$-dimensional simplicial complexes.

**Euler characteristic.** Various geometrical or topological properties for characterising simplicial complexes exist. One such property is the *Euler characteristic*, defined as the alternating sum of the number of simplices in each dimension. For a simplicial complex $K$, we define the Euler Characteristic $\chi$ as

$$\chi(K) = \sum_{n=0}^{\infty} (-1)^n |K^n|, \tag{1}$$

where $|K^n|$ denotes the cardinality of set of $n$-simplices. The Euler characteristic is *invariant* under homeomorphisms and can be related to other properties of $K$. For instance, $\chi(K)$ can be equivalently written as the alternating sum of the *Betti numbers* of $K$. Moreover, the Euler characteristic can be defined for other combinatorial complexes and structures (Robins, 2002).

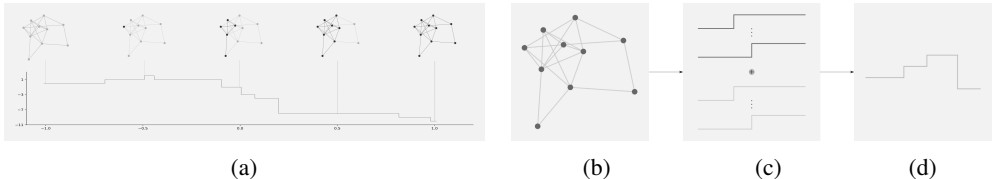

(a)                           (b)          (c)          (d)

Figure 1: The standard algorithm to compute the ECC for a graph, depicted in (a), calculates the vertex filtration heights and sorts them in ascending order. One then loops over each set of predefined height values and keeps a running total of the Euler Characteristic as the number of vertices minus edges with height value less than the current height. Our approach differs in that we calculate the ECC of a graph (b) for each vertex and edge *separately* (c). The sum of the curves is computed for the edges and vertices and the total is subtracted to yield the final ECC (d). **The advantage is a fully parallel computation, making our formulation amenable to hardware accelerations**.

**Filtrations.** The Euler characteristic is limited in the sense that it only characterises a simplicial complex $K$ at a single scale. A multi-scale perspective of this statistic is known to enhance the expressivity of the resulting representations. Specifically, given a simplicial complex $K$ and a function $f \colon \mathbb{R}^n \to \mathbb{R}$, we obtain a multi-scale view on $K$ by considering the function $\tilde{f}$ as the restriction of $f$ to the 0-simplices of $K$, and defining $\tilde{f}(\sigma) := \max_{\tau \subset \sigma} \tilde{f}(\tau)$ for higher-dimensional simplices. With this definition, $\tilde{f}^{-1}((-\infty, r])$ is either empty or a non-empty simplicial subcomplex of $K$; moreover, for $r_1 \leq r_2$, we have $\tilde{f}^{-1}((-\infty, r_1]) \subseteq f^{-1}((-\infty, r_2])$. A function $\tilde{f}$ with such properties is known as a *filter function*, and it induces a *filtration* of $K$ into a sequence of nested subcomplexes, i.e.

$$\emptyset = K_0 \subseteq K_1 \cdots \subseteq K_{m-1} \subseteq K_m = K. \tag{2}$$

Since the filter function was extended to $K$ by calculating the maximum, this is also known as the *sublevel set filtration of $K$ via $f$*.[1] Filter functions (and their induced filtrations) can be learned (Hofer et al., 2020; Horn et al., 2022), or they can be defined based on existing geometrical-topological properties of the input data. Calculating invariants alongside this filtration results in substantial improvements of the predictive power of methods. For instance, calculating homology groups of each $K_i$ leads to *persistent homology*, a shape descriptor for point clouds. However, persistent homology does not exhibit favourable scalability properties, making it hard to gainfully use in practice.

## 4 METHODS

With the *Euler characteristic* being insufficiently expressive and *persistent homology* being infeasible to calculate for large data sets, the *Euler Characteristic Transform* (ECT), created by Turner et al. (2014), aims to strike a balance between the two. Given a simplicial complex $K$ and a filter function $f$,[2] the central idea of the ECT is to compute the Euler characteristic alongside a filtration, thus obtaining a *curve* that serves to characterise a shape (see Fig. 1). If the vertices of $K$ have coordinates in $\mathbb{R}^n$, the ECT is typically calculated based on a parametric filter function of the form

$$\begin{aligned} f \colon S^{n-1} \times \mathbb{R}^n &\to \mathbb{R} \\ (\xi, x) &\mapsto \langle x, \xi \rangle, \end{aligned} \tag{3}$$

where $\xi$ is a *direction* (viewed as a point on a sphere of appropriate dimensionality), and $\langle \cdot, \cdot \rangle$ denotes the standard inner product. For a fixed $\xi$, we write $f_\xi := f(\xi, \cdot)$. Given a *height* $h \in \mathbb{R}$, we obtain a filtration of $K$ by computing the preimage $f_\xi^{-1}((-\infty, h])$. The ECT is then defined as:

$$\begin{aligned} \mathrm{ECT} \colon S^{n-1} \times \mathbb{R} &\to \mathbb{Z} \\ (\xi, h) &\mapsto \chi\left( f_\xi^{-1}\big((-\infty, h]\big) \right), \end{aligned} \tag{4}$$

---

[1] There is also the related concept of a *superlevel set filtration*, proceeding in the opposite direction. The two filtrations are equivalent in the sense that they have the same expressive power.

[2] For notational simplicity, we drop the tilde from the function definition and assume that $f$ constitutes a valid filter function as defined above.

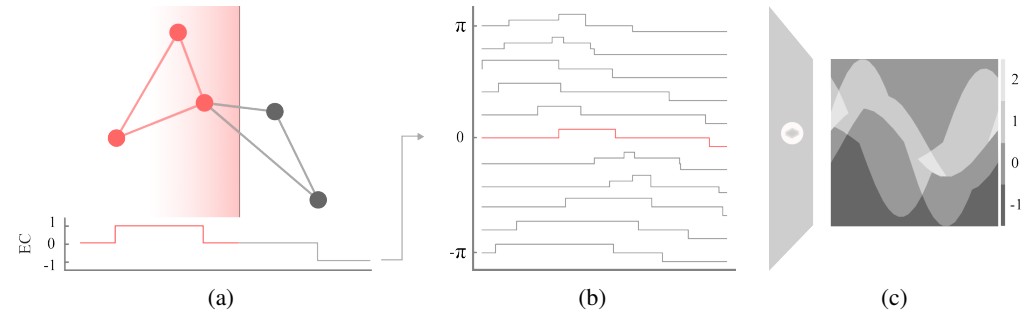

| (a) | (b) | (c) |

Figure 2: Overview of the computation of the Euler Characteristic Transform. (a): Given a graph and a direction, we filter it with a hyperplane (here: from left to right). The nodes and edges of the induced graph are highlighted in red, and the Euler Characteristic Curve of the graph in this direction is displayed below. By the maximum extension principle, edges are added once both target and source node are below the hyperplane. (b): We compute the ECC in multiple directions. The curve in (a) is highlighted in red. On the vertical axis, we parametrise the direction and on the horizontal axis the height. (c): The ECCs are stacked to form an image, where the intensity denotes the Euler Characteristic. This serves as the input for machine learning algorithms.

If $\xi$ is fixed, we also refer to the resulting curve—which is only defined for a single direction—as the *Euler Characteristic Curve* (ECC). The ECT is thus the collection of ECCs calculated from different directions (see Fig. 2). Somewhat surprisingly, it turns out that, given a sufficiently large number of directions $\xi$ (Curry et al., 2022), the ECT is *injective*, i.e. it preserves equality (Ghrist et al., 2018; Turner et al., 2014).

While the injectivity makes the ECT an advantageous shape descriptor, it is currently only used as a static feature descriptor in machine learning applications, relying on a set of pre-defined directions $\xi$, such as directions chosen on a grid. We adopt a novel perspective here, showing how to turn the ECT into a differentiable shape descriptor that affords the integration into deep neural networks, either as a layer or as a loss term. Our **key idea** that permits the ECT to be used in a differentiable setting is the observation that it can be written as

$$\mathrm{ECT}\colon S^{n-1} \times \mathbb{R} \to \mathbb{Z}$$
$$(\xi, h) \mapsto \sum_{k}^{\dim K} (-1)^k \sum_{\sigma_k} \mathbb{1}_{[-\infty, h_\xi(\sigma_k))}(h), \tag{5}$$

where $\sigma_k$ is a $k$-simplex and $h_\xi(\sigma_k)$ is the maximum of the heights in the direction $\xi$ of the vertices that span $\sigma_k$. Eq. (5) rewrites the ECT as an alternating sum of *indicator functions*. To see that this is an equivalent definition, it suffices to note that for the 0-dimensional simplices we indeed get a sum of indicator functions, as the ECT counts how many points are below or above a given hyperplane. This value is also unique, and once a point is included, it will remain included. For the higher-dimensional simplices a similar argument holds. The value of the filter function of a higher-dimensional simplex is fully determined its vertices, and once such a simplex is included by the increasing filter function, it will remain included. This justifies writing the ECT as a sum of indicator functions.

**Differentiability.** A large obstacle towards the development of *topological machine learning* algorithms involves the integration into deep neural networks, with most existing works treating topological information as mere static features. We want our formulation of the ECT to be differentiable with respect to both the *directions* $\xi$ as well as the *coordinates* themselves. However, the indicator function used in Eq. (5) constitutes an obstacle to differentiability. To overcome this, we replace the indicator function with a *sigmoid function*, thus obtaining a smooth approximation to the ECT. Notably, this approximation affords gradient calculations. Using a hyperparameter $\lambda$, we can control the tightness of the approximation, leading to

$$\mathrm{ECT}\colon S^{n-1} \times \mathbb{R} \to \mathbb{Z}$$
$$(\xi, h) \mapsto \sum_{k}^{\dim K} (-1)^k \sum_{\sigma_k} S\left(\lambda \left(h - h_\xi(\sigma_k)\right)\right), \tag{6}$$

where $S(\cdot)$ denotes the sigmoid function. Each of the summands is differentiable with respect to $\xi$, $x_{\sigma_k}$ (the vertex coordinates that span $\sigma_k$), and $h$, thus resulting in a highly-flexible framework for the ECT. We refer to this variant of the ECT as the *Differentiable Euler Characteristic Transform* (DECT). Our novel formulation can be used in different contexts, which we will subsequently analyse in the experimental section. First, Eq. (6) affords a formulation as a shape descriptor layer, thus enabling representation learning on different domains and making a model 'topology-aware.' Second, since Eq. (6) is differentiable with respect to the input coordinates, we can use it to create *loss terms* and, more generally, optimise point clouds to satisfy certain topological constraints. In contrast to existing works that describe topology-based losses (Gabrielsson et al., 2020; Moor et al., 2020; Trofimov et al., 2023; Vandaele et al., 2022), our formulation is highly scalable without requiring subsampling strategies or any form of discretisation in terms of $\xi$ (Nadimpalli et al., 2023).

**Integration into deep neural networks.** Next to being differentiable, our formulation also lends itself to a better integration into deep neural networks. Traditionally, methods that employ ECTs for classification concatenate the ECCs for different directions into a *single* vector, which is subsequently used as the input for standard classification algorithms, after having been subjected to dimensionality reduction (Amézquita et al., 2021; Jiang et al., 2020). However, we find that discarding the directionality information like this results in a loss of crucial information. Moreover, the concatenation of the ECCs requires the dimensionality reduction techniques to be block permutation invariant, as reordering the ECCs should *not* change the output of the classification. This aspect is ignored in practice, thus losing the interpretability of the resulting representation. By contrast, we aim to make the integration of our variant of the ECT *invariant* with respect to reordering individual curves. Instead of using a static dimensionality reduction method, we use an MLP to obtain a learnable embedding of individual Euler Characteristic Curves into a high-dimensional space. This embedding is permutation-equivariant by definition. To obtain a permutation-invariant representation, we use a *pooling layer*, similar to the *deep sets* architecture (Zaheer et al., 2017). Finally, we use a simple classification network based on another MLP. We note that most topological machine learning architectures require a simplicial complex with additional connectivity information to work. This usually requires additional hyperparameters or, in the case of persistent homology, a sequence of simplicial complexes encoding the data at multiple scales. Other deep learning methods, such as deep sets, require a restriction on the number of points in each sample in the dataset. By contrast, our method can *directly* work with point clouds, exhibiting no restrictions in terms of the number of points in each object nor any restrictions concerning the type of sample connectivity information. Hence, DECT can handle data consisting of a mixture of point clouds, graphs, or meshes *simultaneously*.

**Computational efficiency and implementation.** While there are already efficient algorithms for the computation of the ECT for certain data modalities, like image and voxel data (Wang et al., 2022), our method constitutes the first description of a differentiable variant of the ECT in general machine learning settings. Our method is applicable to point clouds, graphs, and meshes. To show its computational efficiency, we provide a brief overview on how to implement Eq. (6) in practice:

1. We first calculate the inner product of all coordinates with each of the directions, i.e. with each of the coordinates from $S^{n-1}$.
2. We extend these inner products to a valid filter function by calculating a *sublevel set filtration*.
3. We translate all indicator functions by the respective filtration value and sample them on a regular grid in the range of the sigmoid function, i.e. in $[-1, 1]$. This is equivalent to evaluating $\mathbb{1}_{[h_\xi(\sigma_k), 1]}$ on the interval $[-1, 1]$.
4. Finally, we add all the indicator functions, weighted by $\pm 1$ depending on the dimension, to obtain the ECT.

All these computations can be *vectorised* and executed in parallel, making our reformulation highly scalable and benefit from GPU parallelism.[3]

## 5 EXPERIMENTS

Having described a novel, differentiable variant of the *Euler Characteristic Transform* (ECT), we conduct a comprehensive suite of experiments to explore and assess its properties. First and

---

[3]Our code is publicly available under `https://github.com/aidos-lab/DECT`.

foremost, building on the intuition of the ECT being a universal shape descriptor, we are interested in understanding how well ECT-based models perform across *different* types of data, such as point clouds, graphs, and meshes. Moreover, while recent work has proven theoretical bounds on the number of directions required to unique classify a shape (i.e. the number of directions required to guarantee injectivity) via the ECT (Curry et al., 2022), we strive to provide practical insights into how well classification accuracy depends on the number of directions used to calculate the ECT. Finally, we also show how to use the ECT to *transform* point clouds, taking on the role of additional optimisation objectives that permit us to adjust point clouds based on a target ECT.

**Preprocessing and experimental setup.** We preprocess all data sets so that their vertex coordinates have at most unit norm. We also centre vertex coordinates at the origin. This scale normalisation simplifies the calculating of ECTs and enables us to use simpler implementations. Moreover, given the different cardinalities and modalities of the data, we slightly adjust our training procedures accordingly. We split data sets following an $80\%/20\%$ train/test split, reserving another $20\%$ of the training data for validation. For the graph classification, we set the maximum number of epochs to $100$. We use the ADAM optimiser with a starting learning rate of $0.001$. As a loss term, we either use *categorical cross entropy* for classification or the *mean squared error* (MSE) for optimising point clouds and directions.

**Architectures.** We showcase the flexibility of DECT by integrating it into different architectures. Our architectures are kept purposefully *simple* and do not make use of concepts like attention, batch normalisation, or weight decay. For the synthetic data sets, we add DECT as the first layer of an MLP with 3 hidden layers. For graph classification tasks, we also use DECT as the first layer, followed by two convolutional layers, and an MLP with 3 hidden layers for classification. By default, we use 16 different directions for the calculation of the ECT and discretise each curve into 16 steps. This results in a $16 \times 16$ 'image' for each input data set. When using convolutional layers, our first convolutional layer has 8 channels, followed by a layer with 16 channels, which is subsequently followed by a pooling layer. Our *classification network* is an MLP with 25 hidden units per layer and 3 layers in total. Since we represent each graph as a $16 \times 16$ image the number of parameters is always constant in our model, ignoring the variation in the dimension of the nodes across the different datasets. We find that this makes the model highly scalable.

## 5.1 LEARNING DIRECTIONS FOR CLASSIFICATION

Table 1: By learning directions, DECT increases accuracy and decreases variance.

| ECT + MLP | |
|---|---|
| Fixed | $77.61 \pm 7.98$ |
| Learnable | $81.29 \pm 3.39$ |

As a motivating example, we study how learning directions affects the classification abilities of DECT. We use the MNIST dataset with each non-zero pixel viewed as a point in a point cloud. For this experiment we use the MLP model and limit the number of directions to 2 instead of 16, chosen uniformly on the unit circle. In the first experiment we keep the directions fixed and in the second we allow DECT to learn the optimal directions. Both models are trained for 10 epochs and the experiment is repeated 10 times. Table 1 depicts the results and it shows that learning directions allows the model to improve the classification accuracy under sparse conditions. For the ECT to be injective in 2D, the minimum number of directions needed is 3 when the vertex coordinates of the dataset are known. When the vertex coordinates are not known in advance, the minimum number of directions depends on the cardinality of the point cloud. The experiment shows that expressivity is not limited when the number of directions is much lower than both the theoretical number of directions needed for injectivity and the cardinality of the point cloud.

## 5.2 OPTIMISING EULER CHARACTERISTIC TRANSFORMS

Our method also lends itself to be used in an optimisation setting. In contrast to prior work (Carrière et al., 2021; Gabrielsson et al., 2020; Moor et al., 2020), representations learned by DECT permit better *interpretability* since one can analyse which directions are used for the classification. The learned directions provide valuable insights into the complexity of the data, highlighting symmetries.

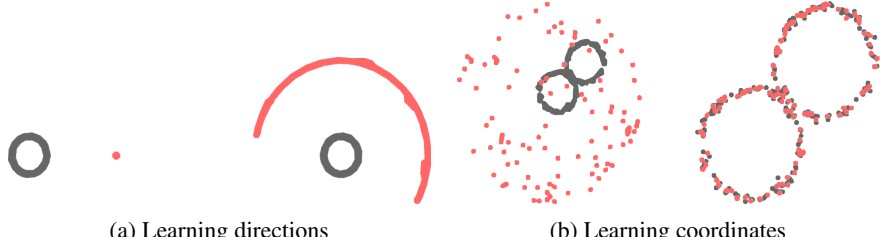

|                                   |                                   |
|-----------------------------------|-----------------------------------|
| (a) Learning directions           | (b) Learning coordinates          |

Figure 3: (a): We sample a noisy point cloud from a circle (grey). Red dots show the directions, i.e. *angles*, used for the ECT (left: initial, right: after training). Our method DECT spreads directions properly over the unit circle, resulting in a perfect matching of the ground truth. (b): DECT also permits us to optimise existing point clouds to match a target ECT in an end-to-end differentiable fashion. Using two point clouds (grey: target; red: input data), we train DECT with an MSE loss between the learned ECT and the target ECT. Starting from a randomly-initialised point cloud (left), point coordinates are optimised to match the desired shape (right). Notably, this optimisation *only* involves the ECT, demonstrating its capabilities as a universal shape descriptor.

**Learning and visualising directions.** We fix a noisy point cloud sampled from a circle, computing the full 'ground truth' ECT with respect to a set of directions sampled uniformly from $S^1$. We then initialise our method DECT with a set of directions set to a random point on the unit circle. Using an MSE loss between the ground truth ECT and the ECT used in our model, we may *learn* appropriate directions. Fig. 3a shows the results of the training process. We observe two phenomena: first, due to the symmetry of the ECT, it suffices to only cover half the unit circle in terms of directions; indeed, each vertical slice of the ECT yields an ECC, which can also be obtained by rotation. The same phenomenon occurs, *mutatis mutandis*, when directions are initialised on the other side of the circle: the axis of symmetry runs exactly through the direction closest and furthest from the point cloud, corresponding to the 'maximum' and 'minimum' observed in the sinusoidal wave pattern that is apparent in the ground truth ECT. We observe that the learned directions are not precisely situated on the unit circle but close to it. This is due to DECT not using a spherical constraint, i.e. learned directions are just considered to be points in $\mathbb{R}^2$ as opposed to being angles.[4] However, the optimisation process still forces the directions to converge to the unit circle, underpinning the fact that DECT in fact learns the ECT of objects even if given more degrees of freedom than strictly required.

**Optimising point clouds.** Complementing the previous experiment on ECT-based optimisation, we also show how to use DECT to *optimise* point cloud coordinates to match a desired geometrical-topological descriptor. This type of optimisation can also be seen as an additional *regularisation* based on topological constraints. In contrast to existing work (Moor et al., 2020; Trofimov et al., 2023; Vandaele et al., 2022), our method is computationally highly efficient and does not necessitate the existence of additional simplicial complexes.https://people.math.ethz.ch/ skalisnik/ To use DECT as an optimisation objective, we normalise all ECTs, thus ensuring that they operate on the same order of magnitude for an MSE loss.[5] Being differentiable, DECT permits us to adjust the coordinate positions of the source point cloud as a function of the MSE loss, computed between the ECT of the model and the ECT of the target point cloud. Fig. 3b illustrates that DECT is capable of adjusting coordinates appropriately. Notably, this also permits us to train with different sample sizes, thus creating *sparse approximations* to target point clouds. We leave the approximation of structured objects, such as graphs or simplicial complexes, for future work; the higher complexity of such domains necessitates constructions of auxiliary complexes, which need to be separately studied in terms of differentiability.

## 5.3 CLASSIFYING GEOMETRIC GRAPHS

Moving from point clouds to graphs, we first study the performance of our method on the `MNIST-Superpixel` data set (Dwivedi et al., 2023). This data set, being constructed from

---

[4]We added spherical constraints for all other classification scenarios unless explicitly mentioned otherwise.
[5]This is tantamount to making DECT scale-invariant. We plan on investigating additional invariance and equivariance properties in future work.

Table 2: Comparing DECT with other methods on the `MNIST-Superpixel` data set. We report overall accuracy (↑) and runtime per epoch (↓), highlighting the fact that even on commodity hardware, our method is an order of magnitude faster than the fastest GNN methods, yielding a favourable trade-off between performance, scalability, and accuracy. Accuracy can be further improved by using a complex constructed from the input images. At the cost of increased runtime for processing faces in the complex, our ECT+MLP method is on a par with more complex graph neural networks. Accuracy values and runtimes of all comparison partners are taken from Dwivedi et al. (2023).

| Method | Accuracy | Epoch runtime (s) |
|---|---|---|
| GAT (Veličković et al., 2018) | $95.54 \pm 0.21$ | 42.26 |
| GCN (Kipf & Welling, 2017) | $90.71 \pm 0.22$ | 83.41 |
| GIN (Xu et al., 2019) | $96.49 \pm 0.14$ | 39.22 |
| GraphSage (Hamilton et al., 2017) | $97.31 \pm 0.10$ | 113.12 |
| MLP | $95.34 \pm 0.14$ | 22.74 |
| ECT+CNN (ours) | $93.00 \pm 0.80$ | 4.50 |
| ECT+MLP (ours) | $97.20 \pm 0.10$ | 10.80 |

Table 3: Results of 5 runs on small graph benchmark data sets. Parameter numbers are approximate because the number of classes differs. The high consistency and performance of our method on the 'Letter' data sets is notable.

| | Params. | BZR | COX2 | DHFR | Letter-low | Letter-med | Letter-high |
|---|---|---|---|---|---|---|---|
| GAT | 5K | $80.3 \pm 2.0$ | $79.2 \pm 2.6$ | $72.8 \pm 3.2$ | $90.0 \pm 2.2$ | $63.7 \pm 6.0$ | $43.7 \pm 4.1$ |
| GCN | 5K | $80.5 \pm 2.4$ | $\mathbf{79.4 \pm 1.8}$ | $\mathbf{76.7 \pm 3.8}$ | $81.4 \pm 1.6$ | $62.0 \pm 2.1$ | $43.1 \pm 1.7$ |
| GIN | 9K | $81.7 \pm 4.9$ | $77.9 \pm 2.4$ | $64.7 \pm 8.3$ | $85.0 \pm 0.6$ | $67.1 \pm 2.5$ | $50.9 \pm 3.5$ |
| ECT+CNN (ours) | 4K | $\mathbf{81.8 \pm 3.2}$ | $70.4 \pm 0.9$ | $67.9 \pm 5.0$ | $\mathbf{91.5 \pm 2.1}$ | $\mathbf{76.2 \pm 4.8}$ | $\mathbf{63.8 \pm 6.0}$ |
| ECT+CNN (ours) | 65K | $84.3 \pm 6.1$ | $74.6 \pm 4.5$ | $72.9 \pm 1.6$ | $96.8 \pm 1.2$ | $86.3 \pm 2.0$ | $85.4 \pm 1.3$ |

image data, has a strong underlying geometric component, which we hypothesise our model should be capable of leveraging. Next to the graph version, we also create a meshed variant of the `MNIST-Superpixel` data set, first assigning to each pixel a coordinate in $\mathbb{R}^2$ by regularly sampling the unit square, then setting the vertices in the simplicial complex to be the non-zero pixel coordinates. We then add edges and faces by computing a *Delaunay complex* of the data (the radius of said complex spans the non-zero pixels). The resulting complex captures both the geometry and the topology of the images in the data set. Following this, we classify the data using DECT and other methods, using a CNN architecture for the original data set and an MLP architecture for its meshed version. Notably, we find that our method only requires about 20 epochs for training, after which training is stopped automatically, whereas others methods use more of the allocated training budget of 100 epochs. Table 2 depicts the results; we find that DECT overall exhibits favourable performance given its smaller footprint. Moreover, DECT exhibits performance on a par with competitor methods on the meshed variant of the data set since the presence of higher-order structural elements like faces enables it to leverage geometric properties. Finally, we want to point out computational considerations. The last column of the table shows the runtimes per epoch. Here, DECT outperforms all other approaches by an order of magnitude or more. The reported runtime is the in fact slowest of all our experiments, with most other data sets only taking about a minute for a *full* 100 epochs. We report the values from Dwivedi et al. (2023), noting that the survey uses a single Nvidia 1080Ti (11GB) GPU was used on a cluster, whereas our model was trained on a Nvidia GeForce RTX 3070 Ti (8GB) GPU on a commodity laptop. This underlines the utility of DECT as a fast, efficient classification method.

We also use a version of DECT to classify point clouds. In contrast to prior work (Turner et al., 2014), we do not use (simplicial) complexes, but restrict the ECT to *hyperplanes*, thus merely counting the number of points above or below a given plane. We then classify shapes from `ModelNet40` using 5 runs, sampling either 100 or 1000 points. In the former case, we achieve an accuracy of $74 \pm 0.5$, while in the latter case our accuracy is $77.1 \pm 0.4$. Given the low complexity and high speed of our model, this is a notable result compared to the performance reported by Zaheer et al. (2017), i.e. $82.0 \pm 2.0$ and $87.0 \pm 2.0$, respectively. Moreover, DECT is not restricted to point clouds of a

specific size, and we believe that the performance gap could potentially be closed for models with more pronounced topological features and varying cardinalities.

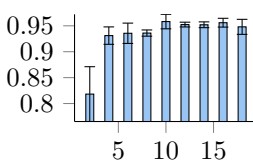

Figure 4: Accuracy on the Letter-low dataset as a function of the number of directions.

As a final experiment, we show the performance of our DECT when it comes to analysing graphs that contain node coordinates. We use several graph benchmark data sets (Morris et al., 2020), with Table 3 depicting the results. Overall, we observe high predictive performance, with DECT outperforming existing graph neural networks while requiring a lower parameter budget. We also show the benefits of substantially increasing the capacity of our model: going to a higher parameter budget yields direct improvements in terms of predictive performance. Interestingly, we observe the highest gains on the 'Letter' data sets, which are subjected to increasingly larger levels of noise. The high performance of our model in this context may point towards better robustness properties, which we aim to investigate in future work. Finally, as Fig. 4 demonstrates, accuracy remains high even when choosing a smaller number of directions for the calculation of the ECT.

## 6 CONCLUSION AND DISCUSSION

We described DECT, the first differentiable framework for *Euler Characteristic Transforms* (ECTs) and showed how to integrate it into deep learning models. Our method is applicable to different data modalities—including point clouds, graphs, and meshes—and we showed its utility in a variety of learning tasks, including both *optimisation* and *classification*. The primary strength of our method is its *flexibility*; it can handle data sets with mixed modalities, containing objects with varying sizes and shapes—we find that few algorithms such adaptability. Moreover, our computation lends itself to high scalability and built-in GPU acceleration; as a result, our ECT-based methods train an order of magnitude faster than existing models on the same hardware. We observe that our method exhibits scalability properties that surpass existing *topological machine learning* algorithms (Hajij et al., 2023; Hensel et al., 2021; Papamarkou et al., 2024). Thus, being fully differentiable, both with respect to the number of directions used for its calculation as well as with respect to the input coordinates of a data set, we extend ECTs to hitherto-unavailable applications.

**Future work.** We believe that this work paves the path towards new future research directions and variants of the ECT. Along these lines, we first aim to extend this framework to encompass variants like the *Weighted Euler Characteristic Transform* (Jiang et al., 2020) or the *Smooth Euler Characteristic Transform* (Crawford et al., 2020). Second, while our experiments already allude to the use of the ECT to solve inverse problems for point clouds, we would like to analyse to what extent our framework can be used to reconstruct graphs, meshes, or higher-order complexes. Given the recent interest in such techniques due to their characteristic geometrical and topological properties (Oudot & Solomon, 2020), we believe that this will constitute a intriguing research direction. Moreover, from the perspective of machine learning, there are numerous improvements possible. For instance, the ECT in its current form is inherently *equivariant* with respect to rotations; finding better classification algorithms that respect this structure would thus be of great interest, potentially leveraging spherical CNNs for improved classification (Cohen et al., 2018). Our experiments on geometric graphs point towards the utility of general geometrical-topological descriptors that offer a complementary approach to established models based on message passing. Leveraging existing approaches based on differential forms (Maggs et al., 2024), we plan on establishing the ECT and its variants as new interpretable methods for general graph learning tasks. Finally, we aim to improve the representational capabilities of the ECT by extending it to address node-level tasks; in this context, topology-based methods have already exhibited favourable predictive performance at the price of limited scalability (Horn et al., 2022). We hope that extensions of DECT may serve to alleviate these issues in the future.

ACKNOWLEDGMENTS

The authors are grateful for helpful comments provided by Jeremy Wayland. They also wish to extend their thanks to the anonymous reviewers and the area chair, who believed in the merits of our work. B.R. is supported by the Bavarian state government with funds from the *Hightech Agenda Bavaria*.

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
