# OpenReview forum: "Differentiable Euler Characteristic Transforms for Shape Classification"
_ICLR.cc/2024/Conference — ICLR 2024 poster_

### Official Review · Reviewer_Y6Ss · 2023-10-31

**Soundness:** 3 good
**Presentation:** 3 good
**Contribution:** 3 good
**Rating:** 6
**Confidence:** 4

**Summary:**

The paper introduces a differentiable version of a geometry descriptor - ECT aka Euler Characteristic Transform, and apply it to the shape classification. In a nutshell, it computes a curve(s) describing a shape (which can be formalized as an exponential family model) as a alternating sum of sigmoids (a "smooth counter" of primitives above / below a collection of hyperplane directions).
Experimental evaluation is provided on a set of benchmarks, and methods performs on-par with a set of GCN-like baselines.

**Strengths:**

- (Clarity) Paper is well-written and is easy to follow, a concise description of the math background will be appreciated by the readers.
- (Originality) Looking at shape descriptors like ECT and studying their performance on realistic applications seems like a novel direction which could benefit a lot of downstream tasks.
- (Significance) Method produces close to state-of-the-art results, and also seems to be quite scalable.
- It looks like the method can also be used when defining loss functions as a way to compare shapes, which is nice.

**Weaknesses:**

- (Novelty, minor) One of the main technical contributions of this work is swapping a Dirac function to a sigmoid with a hyperparameter. It is unclear if this is a non-obvious contribution.
- (Evaluation) Not sure if 5.1 is very meaningful - isn't the task trivial? Providing baseline results would help interpretation.
- (Evaluation) 5.3 - point cloud classification - it is a bit of an overstatement to say that "accuracy of 77%" is "surprising close to" 87.0?

**Questions:**

- How important is the differentiability aspect of DECT? E.g. one could potentially take a predefined set of parameters for (6), and use the output of that as a descriptor? Do you actually estimate the parameters of the transform, and does it add to the performance?
- In Table 2, CNN performs significantly worse than ECT + MLP. A "vanilla" MLP also leads to performance which is quite a bit higher than GCN, which seems strange. Some commentary of reliability of these numbers would be useful.
- (purely out of interest) Would it be possible to combine the proposed description with methods like GCN, e.g. for dense prediction tasks (point cloud segmentation), where descriptor would be per-primitive (e.g. per point).

---

> ### Author Response · Authors · 2023-11-18
>
> We thank the reviewer for the time reviewing our paper. Please find our comments below. We highly appreciate that the reviewer shares our enthusiasm for the novelty of this method.
>
>  > One of the main technical contributions of this work is swapping a Dirac function to a sigmoid with a hyperparameter. It is unclear if this is a non-obvious contribution.
>
> The computation with indicator functions is also novel. One usually does this with a loop, which is less scalable than our method. Moreover, we add differentiability with respect to the angles and with respect to the spatial coordinates, also novel and show our method is sometimes up to an order of magnitude faster than classical GNN methods. When comparing to other topological methods the speedup is even more dramatic.
>
> The application of the ECT to learn directions and do point cloud approximation has not been attempted before, and we believe that this is both a novel and valuable contribution to the literature.
>
> > (purely out of interest) Would it be possible to combine the proposed description with methods like GCN, e.g. for dense prediction tasks (point cloud segmentation), where descriptor would be per-primitive (e.g. per point).
>
> We very much appreciate the curiosity! The DECT, due to its differentiability, is very well suited to be used in a GNN/GCN aggregation layer.  Together with node prediction, we would love to tackle point cloud segmentation in future work. We also believe that we can do this in a flexible and adaptive framework for many types of data.
>
> > How important is the differentiability aspect of DECT? E.g. one could potentially take a predefined set of parameters for (6), and use the output of that as a descriptor? Do you actually estimate the parameters of the transform, and does it add to the performance?
>
> Theoretically yes. There is an upper bound with respect to dimension, the cardinality of the input data and minimum distance between points that guarantees injectivity. This theoretical bound is typically rather high and for point clouds and graphs in practice and it is typically prohibitive to require injectivity. The precise bound can be found in [1], Theorem 7.14.
>
> Hence we use fewer directions and instead opt to learn which directions can discriminate between the classes.
>
> To show that this indeed yields increased classification we ran an additional ablation on the differentiability aspect of DECT. Please see the general remarks above or section 5.1 in the manuscript.
>
> > It looks like the method can also be used when defining loss functions as a way to compare shapes, which is nice.
>
> This is correct and we believe that we can explore this further in future work, hoping to use it for point cloud approximation / sparse approximation and segmentation.

---

### Official Review · Reviewer_5rSB · 2023-11-01

**Soundness:** 3 good
**Presentation:** 2 fair
**Contribution:** 2 fair
**Rating:** 5
**Confidence:** 3

**Summary:**

The paper introduces a novel computational layer called DECT, which enables learning the Euler Characteristic Transform (ECT) in an end-to-end fashion. The ECT is a powerful representation that combines geometrical and topological characteristics of shapes and graphs. The authors overcome the computational limitations of the ECT and demonstrate its scalability and integration into deep neural networks. They also show that DECT achieves competitive performance in graph and point cloud classification tasks.

**Strengths:**

- DECT enables learning the ECT in an end-to-end fashion, overcoming the previous inability to learn task-specific representations.

- The method is highly scalable and can be integrated into deep neural networks as a layer or loss term.

- DECT exhibits advantageous performance in different shape classification tasks for various modalities especially graphs.

**Weaknesses:**

- Although ECT is theoretically injective, it happens only when the number of directions is sufficient. For example, for point cloud classification it could be the case that the number of directions is required to be no less than the cardinality of the point set, for ECT to be injective. This restricts the expressivity, especially for the application on point clouds, and explains why the results on point cloud classification is relatively weaker than graph classification.

- One key contribution of the method is the differentiation on both the coordinates and the directions \ksi. I would like to see an ablation showing the advantage of being able to optimise the direction \ksi, compared to uniformly sampling the direction.

- Number of directions \ksi is set as 16 and I would expect an ablation on different numbers.

-  To apply the method to graph learning, it requires the graph to have spatial coordinates.

**Questions:**

- In Eq 1, (-1)^k should be (-1)^n?

- See weaknesses 2 and 3.

---

> ### Author Response · Authors · 2023-11-18
>
> We thank the reviewer for their time reviewing our paper and for appreciating the utility of DECT for shape classification.
> Please find our comments below.
>
> > Although ECT is theoretically injective, it happens only when the number of directions is sufficient. For example, for point cloud classification it could be the case that the number of directions is required to be no less than the cardinality of the point set, for ECT to be injective. This restricts the expressivity, especially for the application on point clouds, and explains why the results on point cloud classification is relatively weaker than graph classification.
>
> This is indeed true. Guaranteed injectivity depends on the cardinality of the set and dimension see [1]. For classification tasks, only the characteristics of the data distribution are needed, hence we can do with a rather coarse approximation of the ECT and still get useful results. We believe that the point cloud results can also be partially explained by the fact that we eschew an approximation in terms of additional combinatorial complexes. Nevertheless, further research into how to get more expressive representations of the ECT in the context of machine learning are still very much needed.  We would also like to stress that this is the first application of the ECT on more complex datasets, something not attempted yet (except for MNIST and some binary classification tasks) in the literature.
>
> The literature usually applies a coarse dimensionality reduction such as UMAP to the ECT to get a representation for an SVM. Finding better suited classification representations is still an open question. Such a pipeline does not exploit the fact that similar directions share information, and hence organizing them in a way that allows this spatial information in the representation can be exploited by, for instance, a CNN architecture, leads to better performance. The closest analogy is flattening an image to a single vector, whereby one loses the spatial information that a CNN uses to get its performance.
>
> > To apply the method to graph learning, it requires the graph to have spatial coordinates.
>
> We believe that this can be overcome: if data features on the vertices are not present, one can construct them in terms of filtrations, such as node degree or curvature information. Our method will work for these types of metrics as well. Moreover, filtrations can be learned in a task-specific fashion, thus posing no strong restriction of the scope of our work. We will clarify this in the revision.
>
> > Ablations on differentiability and number of directions.
>
> We agree with the reviewer that an ablation on the number of directions would be a valuable addition to the paper.
>
> We provide an ablation study on the number of directions. We find that rather few (compared to theoretical injectivity guarantees) provide significant expressivity.
>
>
>
> [1] 1805.09782.pdf (arxiv.org)

---

### Official Review · Reviewer_v8eE · 2023-11-03

**Soundness:** 2 fair
**Presentation:** 3 good
**Contribution:** 2 fair
**Rating:** 6
**Confidence:** 3

**Summary:**

This paper proposes a differentiable topological shape descriptor that is conceptually based on the Euler characteristic transform (ECT).  Given a shape in its discrete representation as a n-dimensional simplicial complex, ECT computes a descriptor that is a function of a direction and the height function for the topological filtration. The idea is to project the simplices of the shape in all directions and compute the topological property of those directional signatures. Taken together, all these signatures can be concatenated and used together as a global shape descriptor. The main contribution in this paper is to rewrite the ECT in terms of an indicator function (Equation 5), that can be relaxed into a differentiable formulation using the sigmoid, leading to DECT.

The authors have shown a series of experiments to show the benefits of their approach. Table 1 demonstrates a simple proof of concept that DECT classifies shapes of different topologies. Section 5.2 demonstrates how to use it as a loss as well as optimize the descriptor for best directions and section 5.3 for their use in classifying geometric graphs.  Generally, the experiments make a favorable proof of concept.

**Strengths:**

- I find the core idea of this paper to be interesting. Rewriting the topological formulations using more computable components like indicator functions and sigmoids is nice.
- Overall, the paper has been compiled quite well. Despite the relative inaccessibility of the core material, the writing and structure are quite good.
- The choice of experiments to demonstrate the benefits of the descriptor is refreshing. I particularly enjoyed the angle of investigation in sections 5.1 and 5.2, validating the main message of this paper.

**Weaknesses:**

- I find it hard to truly appreciate a more stronger impact of the proposed descriptor for a wide range of applications. Despite the simplicity and comparable accuracies of Table 2, it would be nice to be more direct in explaining what features of data are simply not achievable using standard feature descriptors and how the proposed contributions alleviate it.
- More significantly, I see no baseline comparison with other prior topological descriptors. For eg, how do some of the methods in: (Hajij et al., 2023; Hensel et al., 2021,  (Moor et al., 2020; Trofimov et al., 2023; Vandaele et al., 2022) compare with the proposed construction in section 5.1, 5.2 and 5.3?
- It would be valuable to elaborate more concretely on the multi-scale aspect of the descriptor. I suspect it comes as a result of the height h, but it's hard to easily make this observation in the paper. Please confirm and elaborate.

**Questions:**

- How do you take inner products along given directions for higher dimensional simplices like the edge and face on a mesh?
- The ECT and DECT are a set of descriptors lying on the unit hypersphere, and in higher dimensions working with directions sampled on the unit hypersphere becomes computationally very demanding. How can this be alleviated in the current framework?
- Please annotate/reference the direction and height components in the image of Figure 2, to make it clear.


Overall I am on the border for this paper. On the positive side, the paper has been compiled well and the main idea has been enumerated and experimented as a good proof of concept. However, the lack of comparison with conceptually similar baselines is a strong drawback, and more generally the wider applicability of the method is not promoted well. Taken together, I vote for a borderline accept as a pre-rebuttal rating.

---

> ### Author Response · Authors · 2023-11-18
>
> We thank reviewer v8eE for the review and very much appreciate that the reviewer also finds the core ideas of our work useful!
>
> Please find more detailed responses to your queries below:
>
> > Be more direct in explaining what features of data are simply not achievable using standard feature descriptors and how the proposed contributions alleviate it.
>
> - We believe that our method offers substantial advantages over existing methods along the dimensions of *speed*, *scalability*, *expressivity*, *permutation invariance*, and *flexibility*.
>
> - In terms of *speed*, we are almost an order of magnitude faster than some classical GNN methods. This advantage goes up significantly when comparing to other topological methods. In our framework, training cycles that take regular methods—in particular methods based on message passing—days, take minutes in our framework (and we made sure to use the most optimized implementations of our baseline comparison partners).
>
> - In terms of *scalability*, we are the first topological method that can make use of hardware acceleration and (although not explicitly done in this paper) can also be computed using distributed computing.
>
> - In terms of *expressivity*, our method can also be used in traditional classification tasks and exhibits performance that is on a par with other methods. Notably, we achieve this using a comparatively simple architecture.
> Moreover, our method is *permutation invariant* and can be easily used for sets of arbitrary sizes without requiring additional architectural changes. Finally, in terms of *flexibility*, to our knowledge there is no framework that exhibits this level of versatility. Our method can deal with any type of simplicial complex at the same time without changing anything. For instance, `DeepSets` still assumes a fixed cardinality of the input point clouds. GNNs cannot work directly with point clouds. Neither can `DeepSets` work directly on graphs, or simplicial complexes. Our method works on *all* data types without changing a single line of code.
>
> - Representation: Moreover, it outputs a *fixed* size representation of your data! Hence any standard method can be used afterwards. We display this by considering one of the simplest neural network architectures available and still having decent performance.
>
> - Mathematical guarantees: Viewed as a feature layer, it comes backed up by strong mathematical guarantees. Injectivity being one.
> Moreover, our method can also be used for aggregation layer in a Graph Neural Network architecture. We have not explored this in this paper but plan on doing so in future work.
>
> > Elaborate more concretely on the multi-scale aspect of the descriptor.
>
> For a multi-scale approach, one would have to define a different type of filtration function, in particular an alpha complex would much better capture this type of information. Our method as of now captures spatial information and directional information.
>
> > How do you take inner products along given directions for higher dimensional simplices like the edge and face on a mesh?
>
> Functions on simplicial complexes are defined through an extension principle. One defines a “normal” function on the vertices (in our case the inner product of the vertex coordinates and the direction) and extend it to higher simplices. In our case we use the maximum extension, i.e. a higher-order simplex is assigned the maximum of its vertices. For example, to compute the value of an edge, we take the maximum of the vertex values it is spanned by.
>
> > The ECT and DECT are a set of descriptors lying on the unit hypersphere, and in higher dimensions working with directions sampled on the unit hypersphere becomes computationally very demanding. How can this be alleviated in the current framework?
>
> This is indeed one of the core principles and *advantages* of our method as we do not need to sample uniformly—we are learning directions instead. Since we take inner products with the direction vectors and the coordinates of our n-sphere, computations do not get much more involved and this property is very much in contrast to other topological methods, where the dimensions and size and dimension become prohibitive factors rather quickly. In most cases one computes an alpha or Vietoris-Rips complex and this captures the multi-scales aspect of point clouds in a natural fashion. Comparison with additional topological baselines.
>
> We will discuss the suggested references in more detail, but they deal with different use cases than ours, namely unsupervised representation learning (in particular the ‘Topological Autoencoders’ works and follow-ups). While we believe that the ECT has the potential to be used as a loss term in unsupervised data analysis as well, we plan on postponing such an analysis for future work and believe that additional experiments in that direction would be out of scope. Instead, we restrict ourselves to graph classification and shape analysis.

---

### Official Review · Reviewer_DHr6 · 2023-11-04

**Soundness:** 3 good
**Presentation:** 1 poor
**Contribution:** 3 good
**Rating:** 8
**Confidence:** 3

**Summary:**

The paper describes a differentiable graph descriptor based on Euler Characteristics Curves.  In practice, given a graph and a simplex order k, an ECC is constructed by computing, for each d-simplex, the cosine of the angle of the simplex feature with a predefined direction (filtering function) and counting the number of simplices above a given sequence of thresholds. The idea of the paper is to replace the counting with a sum after a soft thresholding, letting the gradient to be backpropagated to both simplex features and the predefined direction.
The paper shows promising results on graph classification and a proof of concept experiment for pointcloud optimization.

**Strengths:**

The proposed method to compute differentiable ECC is straightforward and consists of simply replacing the counting of the elements above a threshold with a sum after a softmax. Nevertheless, the method has significant potential and allows not only the use of ECCs as a graph descriptor but also to investigate of the most significant direction (it is differentiable w.r.t. the ECC direction) and to ‘invert’ the descriptor and optimize directly the input graph.

**Weaknesses:**

The method description is not easy to follow, and many relevant details are not clear or missing. A detailed list is provided in the question section.
In particular, the architecture description is a bit confusing. In “Integration into deep NN” it is written that MLP + global pooling is used to achieve rotation permutation invariance, but the architecture is then described as a CNN over a 16x16 image. Wouldn’t this break permutation invariance?

A discussion about limits is missing.  For instance, since performing sums over simplices, the method is probably dependent on the sampling density. This is particularly relevant for PC classification. The authors also briefly mention the rotation equivariance of the method, but this is not elaborated much. For instance, if the network is invariant w.r.t. rotation permutation invariance, wouldn’t it make the model also rotation invariant (this probably depends also on the distribution of angles)?

My last concern is about the experimental part. In particular, to prove the importance of optimizing angles, I believe that the paper should compare DECT with building the ECT with fixed angles. Also, the method should be compared with more recent GNN methods, especially based on higher-order simplices. (e.g. Weisfeiler and Lehman Go Topological: Message Passing Simplicial Networks, Provably Powerful Graph Networks)

**Questions:**

- In the case of point clouds, how is the graph built? Do you consider all disconnected points?
- Eq 1 is not clear: what is k in the exponent?
- Notation in eq 3 and 4 is not straightforward, is the second row the actual function definition? What is x?
- I find eq 5 difficult to read, especially the definition of the indicator function 1. Also, what is \sigma_k?
-Table 3 reports 2 times ECT-CNN.

---

> ### Comment · Reviewer_DHr6 · 2023-11-10
> **Sorry for the confusion**
>
> I apologize to the authors. I previously posted the review to another paper. Now it should be ok!

---

> ### Author Response · Authors · 2023-11-18
>
> We thank the reviewer for their helpful comments and positive remarks, especially for recognizing the potential of our method, which we indeed believe to permit copious follow-up work. Concerning the novelty, all computational aspects and the description of the ECT are in fact novel as well and deviate from the existing literature. The **key changes** to existing literature are (1) speed, (2) scalability, and (3) differentiability. Please also check out the resource [1] for additional details. We are more than happy to answer any additional questions you might have!
>
> Please find below a detailed set of responses to the concerns.
>
> > Architecture description: In “Integration into deep NN” it is written that MLP + global pooling is used to achieve rotation permutation invariance, but the architecture is then described as a CNN over a 16x16 image. Break Permutation invariance?
>
> We recognize that additional clarity is required and will rewrite the paper accordingly.  We evaluate two types of architectures in our paper. The first is indeed done by considering the ECT as an image of size {directions} x {discretization steps} and use a CNN or MLP architecture to classify the dataset. The second type of architecture embeds the Euler curve in each direction into a higher-dimensional feature space. We subsequently use the embedding for classification. This means we are not only invariant with respect to permutations of the vertices, but also equivariant with respect to permutations in the directions. We will clarify this in the revision.
>
>  > What is the dependence on the sampling density for point clouds etc.
>
> An exact upper bound for injectivity is given in [2] Theorem 7.14 and depends on the dimension and cardinality of the set. Hence, we believe that our sparse method is rather powerful.
>
> > In the case of point clouds, how is the graph built? Do you consider all disconnected points?
>
> We count the number of points above a hyperplane in a direction theta and let this hyperplane go from $-\infty$ to $\infty$. We do indeed consider all disconnected points as we scale the point cloud to fit in a unit sphere. In the case of point clouds, we do not construct a simplicial complex, we only count the points above hyperplanes in each direction.
>
> > Eq 1 is not clear: what is k in the exponent?
>
> The reviewer caught a typo, it is corrected—thanks! The Euler Characteristic is computed as the alternating sum of the number of simplices in dimension n.
>
> > Notation in eq 3 and 4 is not straightforward, is the second row the actual function definition? What is $x$?
>
> We will clarify this in the document, the $x$ is the spatial coordinate of the vertex and $\xi$ is the direction, viewed as a point on the n-sphere.
>
> > I find eq 5 difficult to read, especially the definition of the indicator function 1. Also, what is $\sigma_k$?
>
> We will add some extra clarification in the manuscript. A k-dimensional simplex is denoted by $\sigma_k$ and $h_\xi(\sigma_k)$ denotes the height of the k-simplex in the direction $\xi$. This new notation is hopefully clearer.
>
> >Table 3 reports 2 times ECT-CNN.
>
> That is a correct observation. In the top row we use a low amount of parameters (4k, see the column next to it) for comparisons with other methods. In the row below it we use a higher number of parameters to see how well the model behaves. Therefore, we have also not considered in the highlighting. We will stress this in the manuscript.
>
> Additional resources:
> - [1] 2307.13940.pdf (arxiv.org)
> - [2] 1805.09782.pdf (arxiv.org)

---

> > ### Comment · Reviewer_DHr6 · 2023-11-21
> >
> > I thank the authors for their response.
> > I still have a couple of questions/observations:
> > - Could they please point me out where in the revised paper they clarified the architecture details and achieved invariances?
> > - Also, it would be nice to insert a statement resuming the results of [2] for set cardinality in the main manuscript. If I'm not wrong, at the moment, it is only referenced for what concerns the minimum number of directions. From the answer, I did not understand if cardinality could be a problem of the current formulation or not (maybe using relative frequencies rather than absolute counts could be a solution in case it is).

---

> > > ### Author Response · Authors · 2023-11-21
> > >
> > > We thank the reviewer for reminding us of the architecture description.
> > >
> > > ### Architecture description
> > > We have updated the manuscript in a clearer formulation and highlighted the changes with respect to the architecture discussion in green, the rest in yellow. The three architectures described and used in the paper are as follows:
> > >   1. First a DECT layer followed by an MLP, through flattening the ECT into a vector. (Denoted as ECT+MLP in the paper)
> > >   2. First a DECT layer followed by a CNN, by treating the ECT as an image. (Denoted as ECT+CNN in the paper)
> > >   3. First a DECT layer followed by an equivariant embedding layer, followed by an MLP network.
> > >
> > > The first two architectures are used in most of the experiments and the third one is only used to classify ModelNet40. The first two are not equivariant with respect to permuting the directions, as the reviewer correctly observed. The third one is invariant to permuting the directions, by viewing the ECC in each direction as a high dimensional point and choosing a permutation invariant layer for the embedding. Note that all architectures are invariant with respect to permuting the labels of the points in the point cloud/graph by construction of the ECT
> > >
> > >  ### Density estimation
> > > Our current understanding is that the reviewer would like to see a study on how variations in cardinality within and amongst classes might affect the classification accuracy.
> > > If point clouds have varying cardinality within each class, this might indeed affect classification accuracy, and this has not been explored. However, any such affects can be mitigated by normalizing the ECT of the point cloud by dividing by the maximum value of the ECT. This is by definition the cardinality, hence normalizing yields an ECT with values between [0,1] in the case of point clouds.
> > > In the case of graphs (or higher dimensional simplicial complexes), the ECT might take negative values so a batch normalization might work better if the different cardinalities indeed negatively affect classification accuracy.
> > >
> > > We agree with the reviewer that an appropriate summary of the above and [2] would benefit the paper. This will require some time and will be addressed in the subsequent revision. If this does not reflect the concern of the reviewer, we would love to hear some more clarification.

---

> > > > ### Comment · Reviewer_DHr6 · 2023-11-22
> > > >
> > > > Thanks for the further clarification, now everything is clear.

---

> > > > > ### Author Response · Authors · 2023-11-22
> > > > >
> > > > > Thanks! Given that today's the last day for changes from our side, we would appreciate it if this was reflected in your score. We believe that the work benefited based on your comments! Is there anything else we can answer or address for you?

---

> > > > > > ### Comment · Reviewer_DHr6 · 2023-11-22
> > > > > >
> > > > > > Thanks, I don't have any further questions. I'll adjust my final score after the discussion period with the PC.

---

> > > > > > > ### Author Response · Authors · 2023-11-22
> > > > > > >
> > > > > > > Thanks, that's much appreciated!

---

### Author Response · Authors · 2023-11-18
**Changes in the updated manuscript**

We would like to thank all reviewers for their time and thoughtful comments. In the revised manuscript we address two key points raised in the reviews:

- Does learning directions increase classification accuracy?
- Do the number of directions limit expressivity when below theoretical guarantees?

In the manuscript an ablation study is added where we address both points. We use only 2 directions for the ECT, which is below the theoretical guarantee for injectivity in an optimal situation where the vertex coordinates are known in advance. When not known, injectivity depends on the cardinality of the point cloud and minimum distance between points.

The accuracy increases from 77.61 for uniformly chosen fixed directions to 81.29 by allowing the model to learn optimal directions. This effect becomes more profound (81.06 to 86.17) when increasing the model capacity to 100 hidden units and increasing the discretization steps to 64. (The result is left out in the manuscript for consistency reasons.)

We are grateful for the reviewers that brought the point up and we see it as a valuable addition.

Some minor changes are listed below:

- Notation change from $f_{\xi}(\sigma_k)$ to $h_{\xi}(\sigma_k)$  in Eq. 5 for better consistency.
- Fixed a typo in Eq. 1 and below Eq. 5
- Equation 3 and 4 changed.

---

### Author Response · Authors · 2023-11-22

Based on the further helpful discussion with the reviewers we have updated the manuscript in the following way. The changes are highlighted in the manuscript.

- We have replaced the overview figure with one that provides a better overview of how the ECT is computed.
- We have rewritten and restructured the "Integration into a deep neural network" to clearer and better reflect our work.
- We have rephrased some sentences.

We hope to have alleviated your concerns. If you find our answers to your comments satisfactory, we would appreciate if this was reflected in your score.

---

### Meta-Review · Area_Chair_nW1g · 2023-12-12

**Metareview:**

The paper proposes a differentiable variant of the Euler Characteristic Transform, which is effective for point cloud and graph classification tasks. The paper was generally positively accepted by the reviewers but there were also critical points raised, including the applicability of the method in cases when the number of direction is not less than the cardinality of the point set, or what happens for graphs without spatial coordinates. Other points raised are the lack of recent comparisons, examples are  (Hajij et al., 2023; Hensel et al., 2021, (Moor et al., 2020; Trofimov et al., 2023; Vandaele et al., 2022) in section 5.1. The authors explain this is out of scope, which is a fair argument since anyways there are no unsupervised experiments in the paper.

**Justification For Why Not Higher Score:**

Good paper, sufficient novelty.

**Justification For Why Not Lower Score:**

The novelty and generality of the method is good.

---

### Decision · Program_Chairs · 2024-01-16

Accept (poster)